# Characterizing the origin of excess dissolved organic carbon in coastal seawater using stable carbon isotope and light absorption characteristics

Heejun Han[1], Jeomshik Hwang[1], Guebuem Kim[1]

[1]School of Earth and Environmental Sciences/Research Institute of Oceanography, Seoul National University, Seoul, 08826, South Korea

*Correspondence to*: Guebuem Kim (gkim@snu.ac.kr)

**Abstract.** In order to determine the origins of dissolved organic matter (DOM) occurring in coastal seawater of the Sihwa Lake, South Korea, which is semi-enclosed artificial lake by a dyke, we measured the stable carbon isotopic ratio of dissolved organic carbon (DOC-$\delta^{13}$C) and optical properties (absorbance and fluorescence) of the DOM in two different seasons (March 2017 and September 2018). The concentrations of DOC were generally higher in lower salinity waters in both periods, while excess of DOC deviating from the mixing line was observed in 2017 in high salinity waters. The main source of DOC in the freshwater-seawater mixing zone was found to be from marine sediments rather than from terrestrial sources based on the DOC-$\delta^{13}$C values (–20.7±1.2‰) and good correlations among DOC, humic-like fluorescent DOM (FDOM$_H$), and NH$_4^+$ concentrations. However, the excess DOC observed in 2017 seems to originate from terrestrial sources by direct land-seawater interactions rather than from in-situ biological production, considering its depleted DOC-$\delta^{13}$C values (–27.8‰ to –22.6‰), higher spectral slope ratio ($S_R$) of light absorbance (indicating low-molecular-weight DOM), and no concurrent increases in FDOM$_H$ and NH$_4^+$ concentrations. The terrestrial DOM source observed in 2017 was found to be non-fluorescent and low-molecular-weight DOM likely because of extended exposure to light and bacterial degradation as this study area is built on the reclaimed land. Our results suggest that the combination of these biogeochemical tools can be a powerful tracer of coastal DOM sources.

## 1 Introduction

Dissolved organic carbon (DOC), a major component of dissolved organic matter (DOM), is the largest reduced carbon pool in the ocean (Benner et al., 1992; Raymond and Spencer, 2014). Understanding of the sources and characteristics of DOC is important since it plays a significant role in coastal carbon dynamics and biogeochemical cycles (Vetter et al., 2007; Carson and Hansell, 2015). In the coastal oceans, DOM sources are diverse including (1) in-situ biological production (Carlson and Hansell, 2015), (2) terrestrial sources such as soils and plant matters (Opsahl and Benner, 1997; Bauer and Bianchi, 2011), and (3) anthropogenic sources including industrial and agricultural wastewater (Tedetti et al., 2010; Griffith and Raymond, 2011).

A part of DOM is known as colored dissolved organic matter (CDOM), which is the light-absorbing fraction of reduced organic matter (Coble, 2007; Kim and Kim, 2016; Kim and Kim, 2018). The major fraction of CDOM, which emits fluorescence after absorbing energy, is referred to as fluorescent DOM (FDOM) (Coble, 2007; Kim and Kim, 2016). Rivers are known as the major source of the humic-like FDOM ($FDOM_H$) in coastal oceans (Stedmon and Nelson, 2015; Kim and

Kim, 2016), while aerobic microbial remineralization of sinking organic matter is the major source of $FDOM_H$ in deep oceans (Jørgensen et al., 2011; Catalá et al., 2015; Kim and Kim, 2016). On the other hand, Kim and Kim (2016) showed that the anaerobic production in the bottom sediments of the deep East Sea (Japan Sea) accounted for about 10% of the total production of $FDOM_H$ in the deep water column. Recently, the anaerobic process in the bottom sediment has been suggested as an important source of $FDOM_H$ in coastal marine environment (Kim and Kim, 2018). In order to obtain such information

on FDOM, an excitation-emission matrix (EEM) spectroscopy method combined with a parallel factor analysis (PARAFAC) model has been employed (Coble, 2007; Kim and Kim, 2016; Kim and Kim 2018). In addition, the absorption spectral slope ratio ($S_R$) can be used as an indicator of molecular weight, source, and photochemistry of DOM since the absorption spectra and spectral parameters for CDOM are largely dependent on its source and photochemical processes (Helms et al., 2008). In general, the $S_R$ values negatively correlate with DOM molecular weight and increase upon irradiation (Helms et al., 2008;

Hansen et al., 2016).

The stable carbon isotopic composition of DOC (DOC-$\delta^{13}$C) has been used to differentiate terrestrial versus marine DOC (Gearing, 1988; Wang et al., 2004; Lee and Kim, 2018; Lee et al., 2020). In general, $\delta^{13}$C values of terrestrial sources such as $C_3$ and $C_4$ plants are in the range of –23‰ to –34‰ and –9‰ to –17‰, respectively, while those derived from marine

phytoplankton are in the range of –18‰ to –22‰ (Gearing, 1998). Since $\delta^{13}$C values for different DOC sources often overlap, the isotope alone is hard to differentiate specific carbon sources. Thus, many previous studies used $\delta^{13}$C values in combination with other parameters such as optical properties and/or DOC:DON (dissolved organic nitrogen) ratios to identify different sources of DOM in coastal environments (Lee and Kim, 2018; Han et al., 2020; Lee et al., 2020).

In this study, we used DOC-$\delta^{13}$C combined with FDOM and $S_R$ values to characterize different sources of DOM in the Sihwa Lake, South Korea, one of the most dynamic coastal settings in terms of salinity changes, hypoxia, metal pollution, and eutrophication (Kim et al., 2009; Ra et al., 2011; Lee et al., 2014; Kim and Kim, 2014; Lee et al., 2017; Kim and Kim, 2018).

## 2 Materials and methods

### 2.1 Study area and sampling

Sihwa Lake (126.6 °E; 37.3 °N) is an artificial seawater lake with an area of 57 km$^2$ and average depth of 3.2 m (maximum depth = 18 m), located on the western coast of South Korea, which was originally constructed as a land reclamation project planned by the government to provide agricultural land and water for the nearby metropolitan area during the 1980s and 1990s (Bae et al., 2010) (Fig. 1). Freshwater runs through the six small streams into the Sihwa Lake and four waterways connect the lake to the Banwol industrial complex (Fig. 1). Since the lake experienced serious deterioration of water quality owing to the wastewater discharge from the industrial complexes under the limited water circulation, the sluice gates were constructed and opened twice a day for the water exchange between the lake and the Yellow Sea since 2012. Then, the dyke is currently used as a tidal power plant (Lee et al., 2017) (Fig. 1). The total volume of the Sihwa Lake water is ~3.3×10$^8$ m$^3$. The discharge rate is approximately 3.4×10$^8$ m$^3$ y$^{-1}$, enough to replace the entire reservoir in a year (Lee et al., 2003; Lee et al., 2014).

Water samples were collected in two different seasons in March 2017 and September 2018. The temperature and salinity were measured using a conductivity-temperature-depth (CTD) instrument (Ocean Seven 304, INDONAUT Srl) onboard a boat (~1 ton). In 2017, sampling was conducted from several depths at all stations. In 2018, only surface water samples were collected at shallow stations (station number 1–6) since the water level of the reservoir was lower than in 2017, and the full depth sampling was conducted at stations 12–14. In order to investigate the effect of industrial wastewater from the industrial complex, an additional sample was collected near the Banwol waterway (station B4) in 2018 (Fig. 1).

Water samples were filtered through a pre-combusted (450 °C for 5h) GF/F filter (pore size = 0.7 μm; Whatman). Samples for DOC and DOC-δ$^{13}$C analyses were acidified with 6M HCl (to a final sample pH of ~2) to avoid any bacterial activities and stored in pre-combusted glass ampoules (Kim and Kim, 2010). Samples for FDOM analysis were stored in pre-combusted amber vials in a refrigerator at 4°C. Samples for dissolved inorganic nutrient analyses were stored frozen in polypropylene conical tubes.

### 2.2 Chemical analyses

Inorganic nutrient concentrations were measured with a nutrient auto-analyzer (QuAAtro39, SEAL analytical). The analytical uncertainties were <5% for the reference materials for NO$_X$ (KANTO, Japan). The dissolved oxygen (DO) concentration was determined using the Winkler's method (Carpenter, 1965). The DOC concentration was measured using a high temperature catalytic oxidation (HTCO) method using a total organic carbon (TOC) analyzer (TOC-V$_{CPH}$, Shimadzu) (Kim and Kim, 2010). Analysis was also conducted for a certified reference material of deep seawater (DSR; 41–45 μM DOC; University of Miami) (Hansell, 2005). The precision of measurement was ±2 μM based on multiple analyses. The

DOC-$\delta^{13}$C values were measured with an isotope ratio mass spectrometer (IRMS; Isoprime, Elementar) connected with a TOC analyzer (Vario TOC cube, Elementar) (Panetta et al., 2008; Troyer et al., 2010). Prior to the analysis, IAEA-CH6 sucrose ($\delta^{13}$C = –10.45±0.03 ‰), Suwannee River Fulvic Acid (SRFA; $\delta^{13}$C = –27.6±0.12 ‰; International Humic Substances Society), and DSR ($\delta^{13}$C = –21.5±0.3 ‰; University of Miami) values were tested to evaluate the accuracy of the measurements (Lang et al., 2007; Panetta et al., 2008; Troyer et al., 2010; Han et al., 2020). Although no certified $\delta^{13}$C value has been reported for DSR, we used the average value reported by Lang et al. ($\delta^{13}$C = –21.7±0.3 ‰) and Panetta et al. ($\delta^{13}$C = –21.4±0.3 ‰).

**2.3 Optical measurements**

Fluorescence and absorbance spectra of the samples were measured using a spectrophotometer (Aqualog, Horiba). For FDOM analyses, the emission and excitation wavelength ranges were set from 240 to 600 nm and from 250 to 500 nm, respectively, with 3 nm scanning intervals (Han et al., 2020). The PARAFAC analysis for the EEM data was performed using the Solo software (Han et al., 2020). The Raman and Rayleigh scattering signals, inner-filter effect, and blank subtraction were corrected using the Solo software (Stedmon and Bro, 2008; Han et al., 2020). The PARAFAC results were validated by a split-half analysis and random initialization (Stedmon and Bro, 2008). The fluorescence intensities of FDOM were normalized with the Raman peak area of water and are presented in Raman Unit (RU) (Lawaetz and Stedmon, 2009).

The PARAFAC model characterized one marine humic-like, one protein-like, and two terrestrial humic-like fluorescent components in Sihwa Lake, which are consistent with previous study (Kim and Kim, 2018) (Fig. S1). The spectral shapes of fluorescent components were compared with previous results from the OpenFluor database (https://openfluor.lablicate.com) (Murphy et al., 2014). All components (C1–C4) were matched with the major components from 36, 39, 62, and 19 studies, respectively, with similarity scores of 95%.

The spectral characteristics of component 1 (FDOM$_C$; Ex/Em = 342/427 nm) and component 3 (FDOM$_A$; Ex/Em = 381/493 nm) are known to be associated with the terrestrial humic-like component originating from terrestrial environment (Coble 2007). Component 2 (FDOM$_M$; Ex/Em = 297/388 nm) is known to be associated with the marine humic-like component originating from microbial remineralization (Coble, 2007; Jørgensen et al., 2011). Component 4 (FDOM$_P$; Ex/Em = 282/322 nm) is characterized as a protein-like (tryptophan-like) component, which originates mainly from biological production (Coble, 2007). In this study, FDOM$_C$ was used as a representative of humic FDOM (FDOM$_H$) since all humic-like components showed a similar pattern.

UV-visible absorption spectra of the samples were measured with a scanning wavelength range of 240–700 nm. The optical indices and parameters of DOM used in this study were prepared as follows. The absorption coefficient was calculated using the following equation:

$$a_\lambda = 2.303 A_\lambda / l \tag{1}$$

where $\alpha$ is the absorption coefficient (m$^{-1}$), $A_\lambda$ is the absorbance, and $l$ is the optical path length of the quartz cuvette (m). The $S_R$ was calculated as the ratio of spectral slope of shorter wavelengths ($S_{275-295}$) to longer wavelengths ($S_{350-400}$) (Helms et al., 2008; Han et al., 2020). The spectral slope ($S$) was calculated using the following equation:

$$a_\lambda = a_{\lambda_{ref}} e^{-S(\lambda - \lambda_{ref})} \tag{2}$$

where $\alpha$ is the Napierian absorption coefficient (m$^{-1}$), $\lambda$ is the wavelength, and $\lambda_{ref}$ is the reference wavelength (Twardowski et al., 2004; Helms et al., 2008).

## 3 Results

In 2017, the vertical distribution of salinity indicated a well-mixed water column (salinity = 28–32) (Fig. 2). Similarly, DO and NH$_4^+$ concentrations were vertically uniform (Fig. 2). The concentrations of DO and NH$_4^+$ were in the ranges of 7–13 mg L$^{-1}$ (average = 10.1±2.4 mg L$^{-1}$) and 0.1–25 μM (average = 8.7±8.1 μM), respectively. However, horizontally, the DO concentration gradually increased with increasing salinity from the innermost station to the outermost station, while the NH$_4^+$ concentration decreased with increasing salinity (Fig. 2). The NH$_4^+$ concentration showed the lowest values (< 1 μM) between station 10 and station 13 (Fig. 2).

In 2018, salinity was in a larger range (salinity = 18–30) compared with that of 2017 (Fig. 2). Especially, low salinity waters (salinity = 18–27) were observed from the innermost station to station 9 (Fig. 2). The concentrations of DO and NH$_4^+$ were in the ranges of 6–11 mg L$^{-1}$ (average = 8.2±1.6 mg L$^{-1}$) and 0.4–25 μM (average = 13.1±7.9 μM), respectively (Fig. 2). The relatively low salinity and DO concentrations were likely associated with the increased freshwater inputs (Fig. 2). The NH$_4^+$ concentrations in the outermost stations were lower than the detection limit (Fig. 2). While the sharp gradients of DO and NH$_4^+$ concentrations were observed at station 9 in 2017, the gradients occurred near station 14 in 2018, associated with the expansion of low-salinity water further to the outer stations (Fig. 2).

In 2017, the vertical distribution of DOC concentrations was quite different from those of salinity and DO concentrations observed in 2018 (Fig. 2). The DOC concentrations were in the range of 97–349 μM (average = 184±76 μM). The highest concentrations of DOC were observed in the surface waters at stations 3, 4, 5, 6, 7, 8, 9 and the bottom waters of stations 3, 4, and 5 (Fig. 2). The DOC-δ$^{13}$C values ranged from –19.2‰ to –27.8‰ (average = –21.8±1.9‰) (Fig. 2). The most depleted DOC-δ$^{13}$C values were found in the surface waters at stations 5, 6, 7, 9, and 10 (–22.6‰ to –27.8‰) (Fig. 2). The concentration of FDOM$_C$ (terrestrial humic-like component 1), FDOM$_A$ (terrestrial humic-like component 2), FDOM$_M$ (marine humic-like component), and FDOM$_P$ (protein-like component) were in the ranges of 1.6–4.1 RU (average = 2.3±0.8 RU), 0.6–1.8 RU (average = 1.1±0.3 RU), 1.0–2.4 RU (average = 1.5±0.5 RU), and 1.6–6.1 RU (average = 2.8±1.0 RU),

respectively (Fig. 3). The concentrations of all FDOM components were generally higher in the upstream stations and decreased with salinity (Fig. 3). The $FDOM_P$ concentration was slightly higher in the bottom water at station 10 (Fig. 3). The $S_R$ values, a proxy for DOM molecular weight, were in the range of 0.70–1.76 (average = 1.21±0.20). Higher $S_R$ values were observed in the surface waters at stations 6, 8, 9, and 10 in 2017 (Fig. 3).

160

In 2018, the concentrations of DOC were in the range of 101–195 μM (average = 130±32 μM). The DOC concentrations gradually decreased with increasing salinity (Fig. 2). The DOC-$\delta^{13}$C values ranged from –19.1‰ to –21.5‰ (average = –20.0±0.6‰) (Fig. 2). The concentrations of $FDOM_C$, $FDOM_A$, $FDOM_M$, and $FDOM_P$ were in the ranges of 1.4–5.1 RU (average = 1.9±0.9 RU), 1.3–4.1 RU (average = 1.8±0.7 RU), 1.4–4.9 RU (average = 2.1±0.9 RU), and 1.1–2.5 RU (average = 1.6±0.4 RU), respectively (Fig. 3). All humic-like FDOM concentrations were higher in 2018 than in 2017 (Fig. 3). The $FDOM_P$ concentrations were generally higher in the surface water and showed a slight increase at station 12 where the salinity is slightly lower (Fig. 3). The $S_R$ values were in the range of 0.72–1.08 (average = 0.87±0.10) (Fig. 3). The $S_R$ values were relatively constant at all sampling stations (Fig. 3).

## 4 Discussion

In both sampling periods, low-salinity waters showed higher DOC, lower DO, higher $NH_4^+$, and higher $FDOM_H$ concentrations (Figs. 2 and 3). As such, the DOC and $FDOM_H$ concentrations exhibited significant correlations against salinity in both periods with different slopes (Fig. 4a and 4b). The DOC concentrations also exhibited good correlations with $NH_4^+$ concentrations in both periods with different slopes, while the $FDOM_H$ concentrations showed a good correlation with $NH_4^+$ concentrations with a single slope (Fig. 5). In coastal region, the excess DOC can be derived from various sources including in-situ biological production, terrestrial source inputs via rivers, and bottom sediment porewater (Hopkinson et al., 1998; Alperin et al., 1999; Kawasaki and Benner, 2006). Our correlation trends suggest the major contribution of DOC either from terrestrial freshwater input or by production in the estuarine mixing zone.

The DOC-$\delta^{13}$C values showed different trends in both sampling periods (Fig. 4c). In 2018, the DOC-$\delta^{13}$C values ranged from –19.1‰ to –21.5‰ (average = –20.0±0.6‰), falling within the range of marine phytoplankton values (–18‰ to –22‰), while the DOC-$\delta^{13}$C values in 2017 were in a larger range from –19.2‰ to –27.8‰, including both marine and terrestrial signatures (Gearing, 1988) (Fig. 4c). The $S_R$ values were relatively low and constant (average = 0.86±0.1) at all stations in 2018, while those exhibited large variations from 0.70 to 1.76 in 2017 (Fig. 3 and 4d). Since $S_R$ values are negatively correlated to molecular weight of DOM and increase on irradiation, such large variations in $S_R$ values in 2017 suggest different history of photodegradation and biological degradation (Moran et al., 2000; Helms et al., 2008).

In both sampling periods, the main source of DOC, dependent on salinity, could be from terrestrial sources as observed in other coastal waters of Korea (Lee et al., 2020). In this study, there was no excess DOC observed in 2018 at station B4 where the waterway connects to the Banwol industrial complex, indicating that anthropogenic source waas insignificant (Fig. 2). However, Kim and Kim (2018) hypothesized that $FDOM_H$ is produced by anaerobic decomposition of organic matter in bottom sediments in the freshwater-seawater mixing zone, based on good correlations among salinity, $NH_4^+$, and $FDOM_H$ concentrations in this lake. Although $FDOM_H$ concentrations showed two different slopes against salinities, a single slope was observed for the correlations against $NH_4^+$ concentrations, indicating a possible main source of $FDOM_H$ associated with $NH_4^+$ productions (Fig. 4b and 5b). Thus, this isotope trend (marine signature), together with the correlations among salinity, DOC, $NH_4^+$, and $FDOM_H$ in both years, suggests that high DOC concentrations occurring in low salinity waters were mainly from marine sediments by anaerobic bacterial production as suggested by Kim and Kim (2018) and that primary terrestrial source of DOC and $FDOM_H$ through original freshwater are insignificant (Fig. 4a, 4b, 5b).

In 2017, the sources of DOC were more complicated, showing significantly higher DOC concentrations and the excess DOC independent of salinity (Fig. 2). Thus, the higher DOC samples observed in 2017 were separated into two groups (Group 1 and Group 2) based on their DOC concentrations, DOC-$\delta^{13}$C values, and salinities (Fig. 2). Group 1 ($n=12$) includes excess DOC samples observed in stations 3, 4, 5, 6, 7, 8, 9, 12, and 13 (Fig. 2). Group 2 ($n=7$) includes excess DOC samples observed in the surface waters of stations 4, 5, 6, 7, 8, 9, and 10 (Fig. 2). Here, the excess denotes the concentrations higher than the salinity mixing line observed in 2018 (Fig. 4a). The excess DOC samples observed in 2017 were ~75% higher than the mixing line (Fig. 4a).

For Group 1 samples, the DOC concentrations ranged from 130 to 330 μM (average = 188±68 μM) (Fig. 2 and 4a). The DOC-$\delta^{13}$C values of Group 1 ranged from −19.2‰ to −23.4‰ (average = −21.3±1.2‰), which are close to the $\delta^{13}$C values of marine organisms (−18‰ to −22‰) (Gearing, 1988) (Fig. 2 and 4c). For this group, $FDOM_H$ concentrations showed on significant increase (Fig. 3 and 4b). Also, $S_R$ values (average = 1.11±0.2) showed relatively lower and constant values than that of Group 2 (Fig. 3 and 4d). The higher DOC concentrations observed in the near bottom waters of stations 3, 4, and 5, in high salinity waters, seem to be from bottom sediment porewater by diffusion process (Koepfler et al., 1993; Alperin et al., 1999; Alkhatib et al., 2013; Kim and Kim, 2018) (Fig. 2). The higher DOC concentrations observed in the deep waters of stations 11, 12, and 13 seem to be from either in-situ biological production or by the transport of bottom sediment sources.

For Group 2 samples, the DOC concentrations ranged from 103 to 291 μM (average = 188±57 μM) (Fig. 2 and 4a). The DOC-$\delta^{13}$C values ranged from −22.6‰ to −27.8‰, which include the signature of of terrestrial $C_3$ plants (−23‰ to −32‰) (Gearing, 1988) (Fig. 2 and 4c). For this group, $FDOM_H$ concentrations showed no significant increases relative to $NH_4^+$ or salinity, indicating excess DOC concentrations were not associated with the common $FDOM_H$ sources observed in both sampling periods (Fig. 3 and 5b). However, $S_R$ values (average = 1.37±0.3) were higher than the other stations likely due to

the influence of low-molecular weight DOM (Helms et al., 2008) (Fig. 4d). Thus, our results suggest that the excess DOC occurring in high-salinity waters in Group 2, which are characterized with non-fluorescent terrestrial sources, were introduced by direct land-seawater interaction through the tidal inundation of seawater on the reclaimed land as this study site is constructed on the reclaimed land (Lee et al., 2020). This may happen if terrestrial DOM (based on DOC-$\delta^{13}$C values) went through intense light exposure (producing non-fluorescent DOM) and/or bacterial degradation on land.

If only salinity and $FDOM_H$ were used to trace the source of the excess DOC occurring in Group 2, in-situ production of DOC by biological production can be simply regarded as a main source since there were no significant changes in these parameters. As such, terrestrial source could be regarded as a main source of the excess DOC occurring in low-salinity waters in Group 1 since there were good correlations between salinities and DOC or $FDOM_H$. Therefore, our study suggests that the combination of carbon stable isotope, $FDOM_H$, and $S_R$ values provides a critical tool to decipher the sources and characteristics of DOM in coastal waters where various DOM sources are present.

## 5 Conclusions

The different sources and distributions of DOM were determined in different seasons using various tracers in the Sihwa Lake, South Korea. Our results revealed that the high DOC concentrations occurring in low-salinity water, which are previously believed to be from terrestrial sources, went from marine sediment sources based on DOC-$\delta^{13}$C values (−21.5‰ to −19.1‰) together with significant correlations among DOC, $FDOM_H$, and $NH_4^+$ concentrations. The high DOC concentrations occurring in high-salinity waters, which are generally believed to be from marine sources, were found to be from non-fluorescent, low-molecular-weight, terrestrial DOM sources based on depleted DOC-$\delta^{13}$C values (−22.6‰ to −27.8‰) and higher $S_R$ values (1.37±0.3), without concurrent increases in $FDOM_H$ and $NH_4^+$ concentrations. Our results demonstrate possibility that the combination of these multiple DOM tracers can be used successfully in other coastal waters where the sources and characteristics of DOM are complicated.

*Data availability.* All data are available upon request to the corresponding author.

*Author contributions.* GK contributed to the conceptualization of the manuscript. HH and GK were involved in planning the research. HH collected samples and performed the analyses. All authors were involved in analyzing the results and writing the paper.

*Competing interests.* The authors declare that they have no conflict of interest.

*Acknowledgements.* This research was supported by the National Research Foundation (NRF) of Korea (NRF-255 2018R1A2B3001147) funded by the South Korean government. We would like to thank all lab members for their assistance.

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

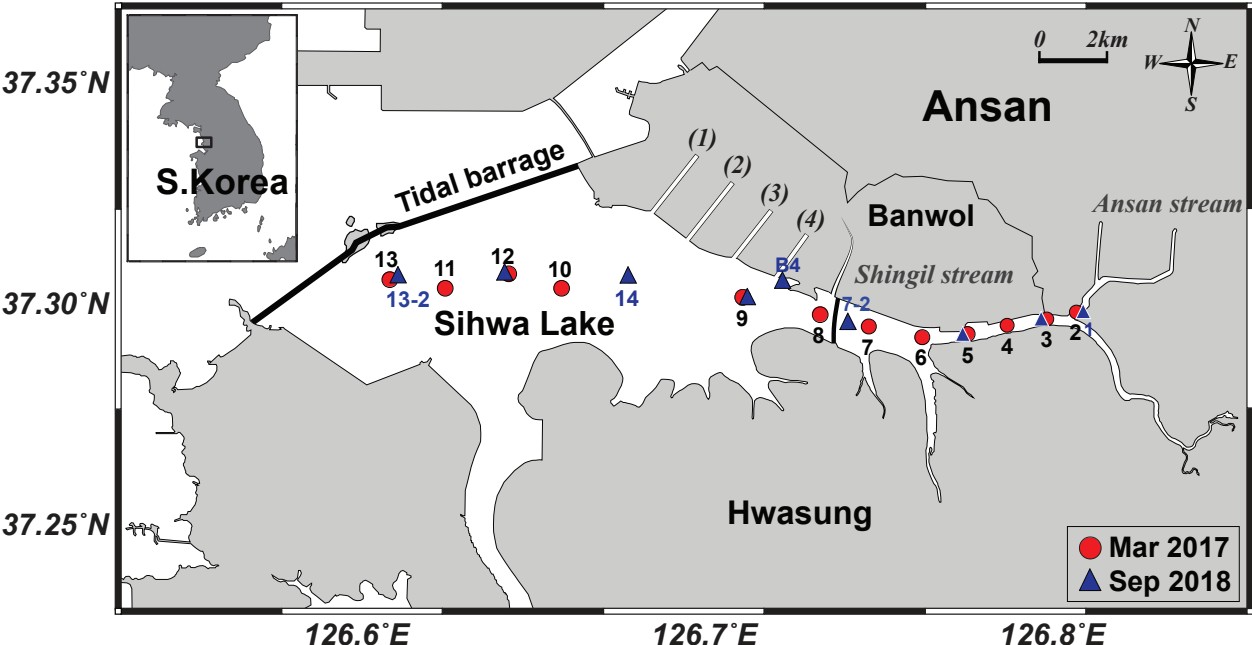

**Figure 1: Map of sampling stations in Sihwa Lake, South Korea.**


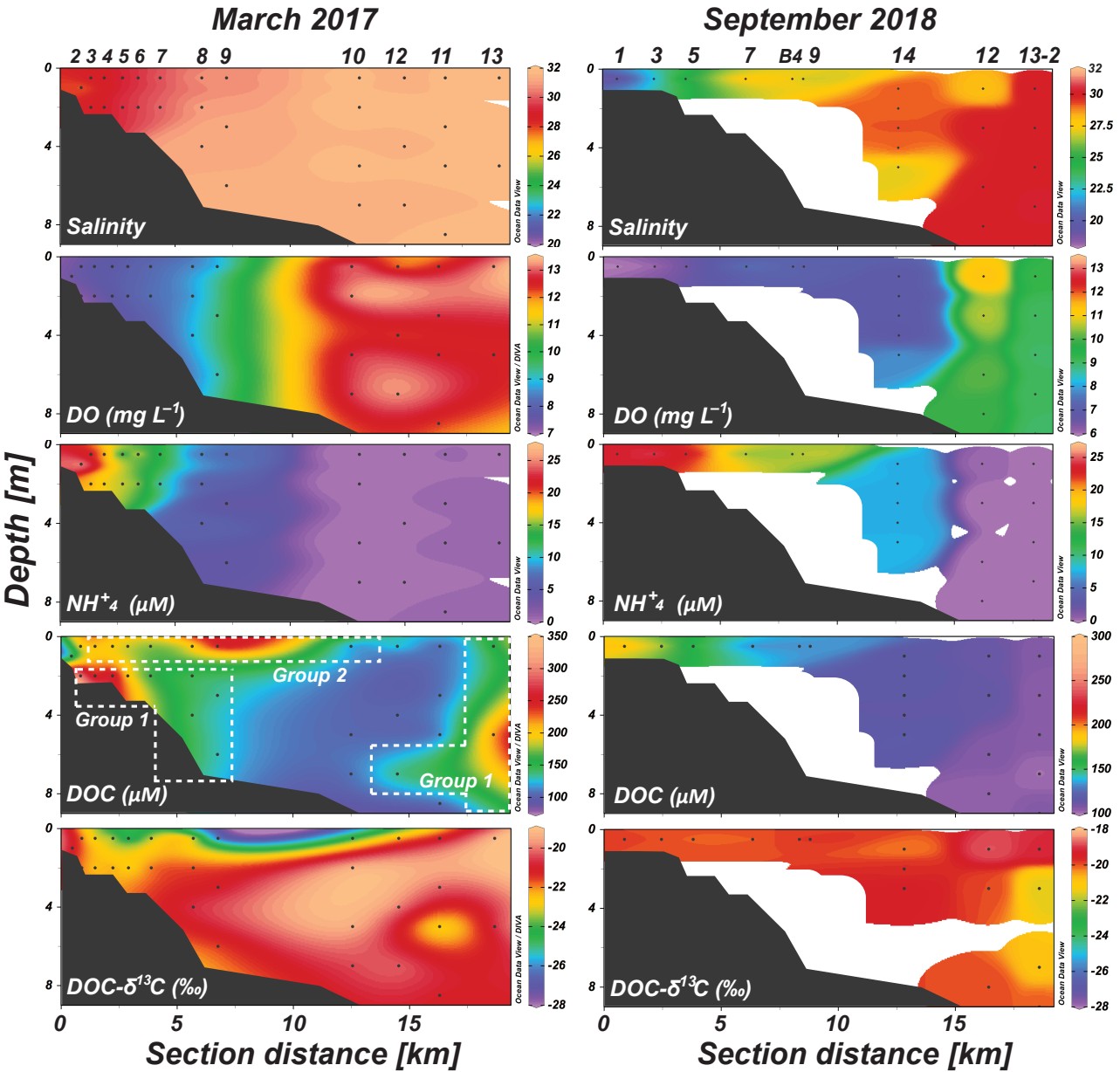

**Figure 2: Vertical distributions of salinity, DO, NH₄⁺, DOC concentrations, and DOC-δ¹³C values in Sihwa Lake in March 2017 and September 2018. The dashed blocks represent stations belonging to Group 1 and Group 2.**


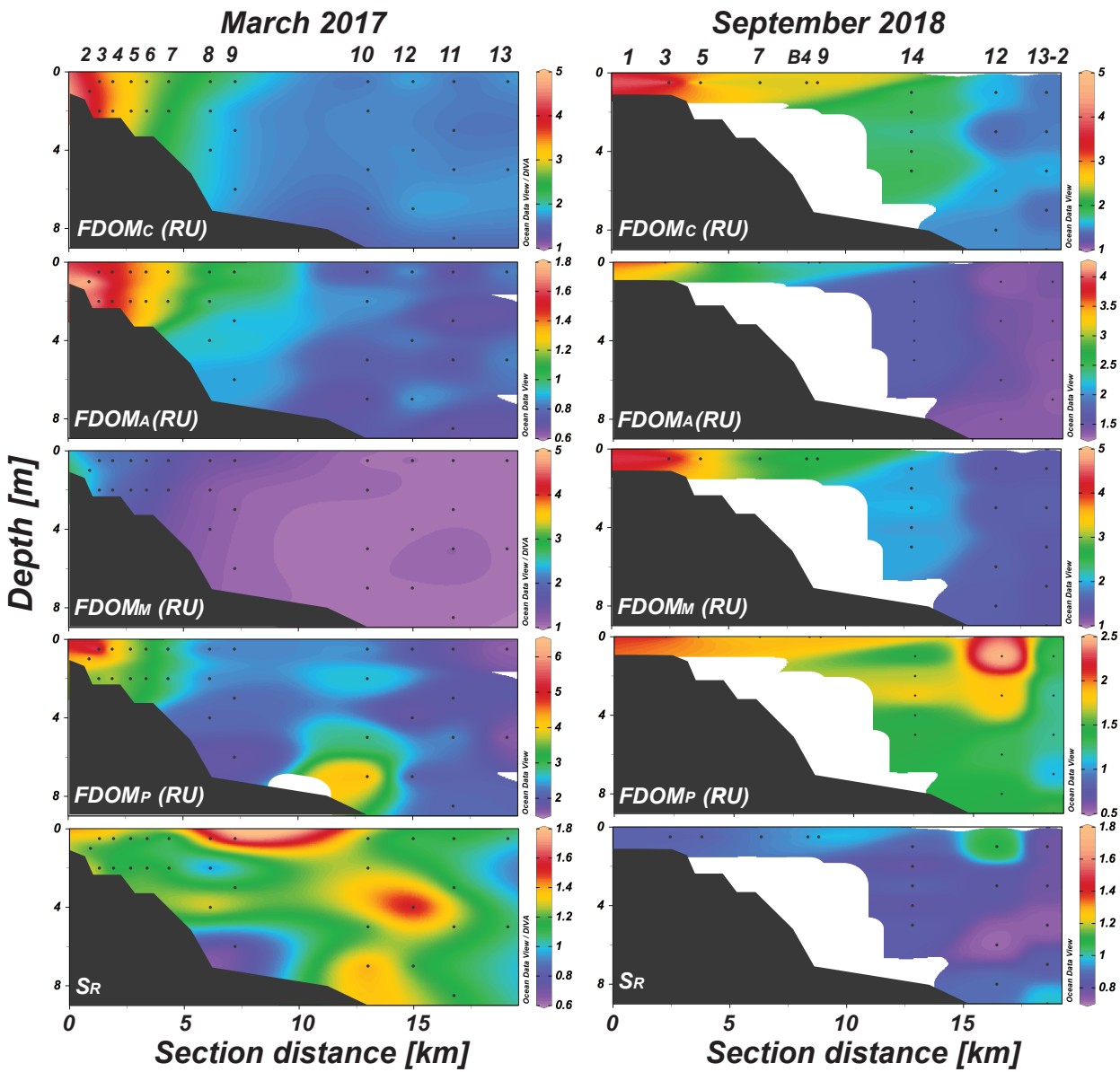


**Figure 3: Vertical distributions of FDOM$_C$, FDOM$_A$, FDOM$_M$, FDOM$_P$, and S$_R$ values in Sihwa in March 2017 and September 2018.**


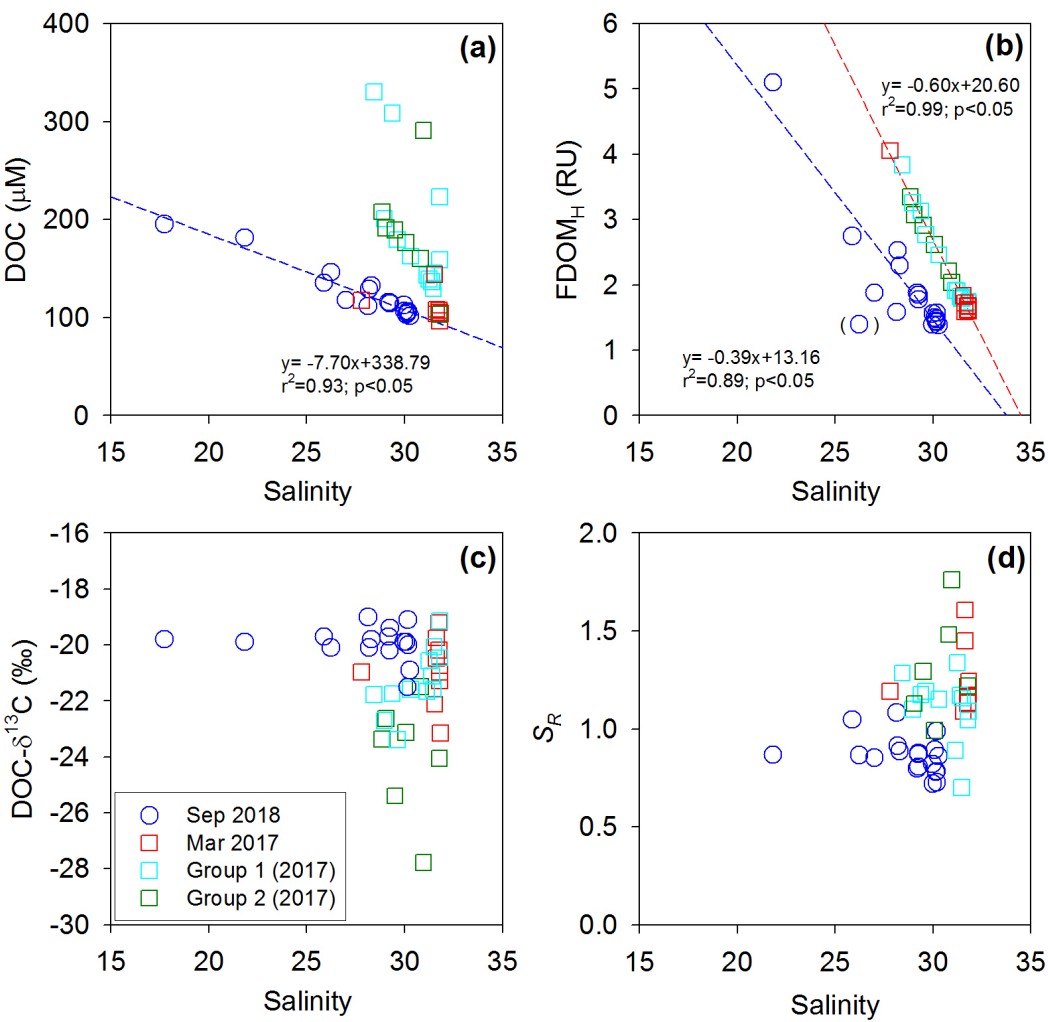


**Figure 4: Plots of the salinity versus (a) DOC concentrations, (b) FDOM$_H$ concentrations, (c) DOC-$\delta^{13}$C values, and (d) $S_R$ values in Sihwa Lake in March 2017 (red square) and September 2018 (blue circle). The dashed lines represent the regression lines.**



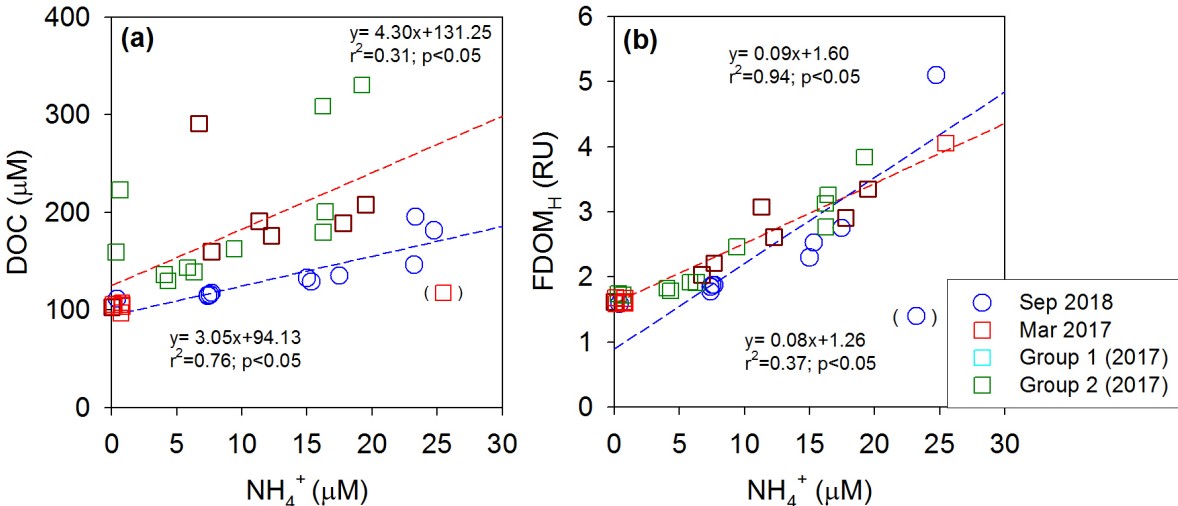

**Figure 5: Plots of the NH$_4^+$ concentrations versus (a) DOC concentrations and (b) FDOM$_H$ concentrations in Sihwa Lake in March**
**2017 (red square) and September 2018 (blue circle). The dashed lines represent the regression lines.**