# Peer review of "Characterizing the origin of excess dissolved organic carbon in coastal seawater using stable carbon isotope and light absorption characteristics"

_Biogeosciences, 2020_

## Referee Comment (RC1) · Anonymous Referee #1 · 29 Aug 2020

This is, nothing really novel, but a simple and clear study. The authors used combined concentration, stable carbon isotope and fluorescence measurements to characterize the sources of dissolved organic carbon (DOC) in a small coastal bay, the Sihwa Lake in South Korea. The manuscript is well written and the results were clearly described and discussed. I support its publication and wish the following suggestions can be considered.

1. The studied Sihwa Lake is a very shallow (<8 m) coastal bay and it appears that the freshwater influence from the few streams to the bay is insignificant from the salinity and DOC-ïĄď13C distributions, only in the land-ocean interface. Therefore, the sediment resuspension could be an important factor influence the DOC concentrations in the bay. The high DOC (or excess DOC) observed in the bottom water in the nearshore stations (only 2-3 meters deep) in 2017 could be influenced by the sediment resuspension or disturbance of the surface sediment, resulting in high sediment porewater DOC fluxed into the bottom water. Usually, the concentrations of porewater DOC in the coastal sediments are much higher than that of the water column. I think the authors mentioned this but more discussion will be good. 2. Line 61-62, "The total volume of the Sihwa Lake water is $\sim$3.3 x 108 m3 y-1 and the discharge rate is approximately 3.4 x 108 m3 y-1 (Lee et al., 2014)". Please check on the unit of the total volume. 3. For Figure 4 and 5, if these lines are linear regression of the date, the regression parameters should be given. 4. In all figure captions, Lake should be added after Sihwa.

---

## Author Comment (AC1) · 3 Sep 2020

Reviewer # 1

General comments: This is, nothing really novel, but a simple and clear study. The authors used combined concentration, stable carbon isotope and fluorescence measurements to characterize the sources of dissolved organic carbon (DOC) in a small coastal bay, the Sihwa Lake in South Korea. The manuscript is well written and the results were clearly described and discussed. I support its publication and wish the following suggestions can be considered.

[Figure]

-> Thank you for your valuable comment. All your comments are carefully taken into account in the revised version.

Specific comments: 1. The studied Sihwa Lake is a very shallow (<8 m) coastal bay and it appears that the freshwater influence from the few streams to the bay is insignificant from the salinity and DOC-$\delta$13C distributions, only in the land-ocean interface. Therefore, the sediment resuspension could be an important factor influence the DOC concentrations in the bay. The high DOC (or excess DOC) observed in the bottom water in the nearshore stations (only 2–3 meters deep) in 2017 could be influenced by the sediment resuspension or disturbance of the surface sediment, resulting in high sediment porewater DOC fluxed into the bottom water. Usually, the concentrations of porewater DOC in the coastal sediments are much higher than that of the water column. I think the authors mentioned this but more discussion will be good.

-> We agree that the excess DOC could be from the sediment re-suspension or porewater exchange . We add more in-depth discussion on this based on DOC-$\delta$13C values (average: –22.1‰ in these samples and references in the revised version.

2. Line 61–62: "The total volume of the Sihwa Lake water is $\sim 3.3 \times 108$ m3 y-1 and the discharge rate is approximately $3.4 \times 108$ m3 y-1 (Lee et al., 2014)". Please check on the unit of the total volume.

-> Yes, corrected. Thank you!

3. For Figure 4 and 5, if these lines are linear regression of the date, the regression parameter should be given.

-> We add the statistical information (r2 and p values) in the revised version.

4. In all figure captions, Lake should be added after Sihwa.

-> Yes, add "lake" as suggested in the revised version. [END]

---

## Referee Comment (RC2) · Anonymous Referee #2 · 6 Sep 2020

Han et al provide a short summary of DOM properties in Sihwa Lake, a constructed coastal lake in a heavily industrialized coastal area, over 2 sampling trips taken in spring 2017 and in late summer 2018. Same sites were visited in each sampling. Using a combination of nutrients and optical and stable isotope tracers, they aim to distinguish multiple sources of DOM (though the sources are not clearly identified). The brevity of this manuscript makes it very difficult to follow. Many details are lacking and some deeper analysis is required to support the conclusions made in this study. Several conclusive statements are made without a clear logical argument to help the reader reach the same conclusion. These problems occur throughout this version of the manuscript, and, along with some substantial editing for grammar and usage, re-

quire more than substantial revision. Specific comments L55: Finish the set up for this manuscript. What are the sources expected? It is curious why the authors didn't try to use endmember mixing analysis (EMMA) to disentangle the sources. The primary sources appear to be: terrestrial, marine, phytoplankton, and "anaerobic benthic processes" which I shorten to benthic. Methods L68: It appears the sluice gates are mostly closed; what does periodic openings entail? Were the gates opened prior to sampling? L70 What vessel was used for sampling? "a ship" is nebulous. L78 How many mL of 6 M HCl were used and what was the final pH? L88 Unlikely that the precision of the TOC analyzer for DSR measurement is 2.2 $\mu$M, round to 2 $\mu$M. How many analyses? L92: To my knowledge no consensus value of DSR is reported, though similar values have been reported as described here. Reword to indicate this (as was done in earlier work referenced here). If a consensus value is now published, please cite the publication. Also please report number of analyses (N) for these standards. Results Make the colorbar ranges for Figs 2 and 3 the same for each panel for ease of comparison. The PARAFAC results should be tested against the OpenFluor database. Here, spectral components are compared to Coble 2007 wherein peaks are visually identified. Suprisingly, the authors only describe 2 of the 4 peaks they find with the model. I recommend they discuss the dynamics of the protein-like component. Given the results presented, it would be informative to see how well this component correlates to other PARAFAC components and in cross section across the lake (ie, as in Figs 2 and 3). Also, correlation of this peak with $\delta$13C values. L140 Range of values does not capture the most negative value reported (-27.8‰ Discussion L155: How is "significant excess" being defined? It is unclear what the authors mean by this phrase and how they quantified it. L158: What does land-seawater interaction mean? Mixing? Proportional mixing would not add an excess of DOC; an excess implies production in spite of mixing. . . unless a 3rd source is implied. In this case, binary mixing analysis won't work. Perhaps the authors should suggest here the benthos as a potential source; but that source also should be parameterized (eg, what is its $\delta$13C-DOC value, FDOMH, FDOMM values, SR, etc.) L180: The groupings appear arbitrary; what criteria were used to separate them? I don't understand how the terrestrial source of DOM can be not fluorescent, given that the authors identify humic fluorescence as a specific marker. This section of the discussion is extremely hard to follow. L197: No evidence is provided for photochemical or bacterial degradation in this study. L201: As suggested earlier, the possibility to use EMMA or other multivariate means with these data are encouraging. I recommend the authors try to analyze their results with an aim of using exploratory methods (eg ordination such as PCA or non-parametric techniques) and perhaps 2-way analyses wherein the difference of season (or stream flow if available; not presented) is considered. A clearer way of quantifying the Groups (1 and 2) must be presented at the very least, so that readers can follow the study. L210: No analysis was presented to demonstrate the linkage of $\delta$13C values and NH4+ values

---

## Author Comment (AC2) · 11 Sep 2020

Reviewer #2

General comments: Han et al provide a short summary of DOM properties in Sihwa Lake, a constructed coastal lake in a heavily industrialized coastal area, over 2 sampling trips taken in spring 2017 and in late summer 2018. Same sites were visited in each sampling. Using a combination of nutrients and optical and stable isotope tracers, they aim to distinguish multiple sources of DOM (though the sources are not clearly identified). The brevity of this manuscript makes it very difficult to follow. Many details are lacking and some deeper analysis is required to support the conclusions made in

this study. Several conclusive statements are made without a clear logical argument to help the reader reach the same conclusion. These problems occur throughout this version of the manuscript, and, along with some substantial editing for grammar and usage, require more than substantial revision.

-> Thank you for your review and comments. In this study, we focused on determining the sources of excess DOC occurring in this bay. Although we cannot elucidate the exact sources and processes in this study, it is clear that our approach (using DOC–$\delta$13C and optical properties) suggests different sources that cannot be identified with the traditional methods. We showed that the excess DOC dependent on salinity is from marine sources (although it is generally regarded as terrestrial sources), and the excess DOC in the high-salinity water is from terrestrial sources (although it is believed to be from marine sources without our approach). The revised version was thoroughly checked for grammar and usage by a native editor.

Specific comments:

1. L55: Finish the set up for this manuscript. What are the sources expected? It is curious why the authors didn't try to use endmember mixing analysis (EMMA) to disentangle the sources. The primary sources appear to be: terrestrial, marine, phytoplankton, and "anaerobic benthic processes" which I shorten to benthic.

-> The end-member mixing analysis is very useful for tracing different water-mass mixings. However, the excess DOC occurring in this study is either from the sediment or land as the DOC is directly introduced to low-salinity water or seawater. Thus, it is impossible to do EMMA.

2. Methods L68: It appears the sluice gates are mostly closed; what does periodic opening entail? Were the gates opened prior to sampling?

-> The sluice gates are opened twice a day (every low tide and high tide). Samplings were conducted in between the openings. We added more details in the revised version.

3. L70: What vessel was used for sampling? " a ship" is nebulous.

-> It is a small boat (∼1 ton). mentioned in the revised version.

4. L78: How many mL of 6M HCl were used and what was the final pH?

-> We added 20 $\mu$L of 6M HCl to each sample. The final pH of the sample was ∼2. Details are described in the revised version.

5. L88: Unlikely that the precision of the TOC analyzer for DSR measurement is 2.2 $\mu$M, round to 2 $\mu$M. How many analyses?

-> corrected as suggested in the revised version. We measured DSR three times per each 10 sample batch.

6. L92: To my knowledge no consensus value of DSR is reported, though similar values have been reported as described here. Reword to indicate this (as was done in earlier work referenced here). If a consensus value is now published, please cite the publication. Also appear report number of analyses (N) for these standards.

-> The $\delta$13C values for the DSR were reported by Panetta et al. (2008) (–21.37±0.33‰ ) and Lang et al. (2007) (–21.9±1.3‰ ). We added more references in the revised version.

7. Results: Make the colorbar ranges for Figs 2 and 3 the same for each panel for ease of comparison.

-> changed as suggested in the revised version.

8. The PARAFAC results should be tested against the OpenFluor database.

-> Yes, the PARAFAC results are compared with the OpenFluor database. We added the results in the revised version.

9. Here, spectral components are compared to Coble 2007 wherein peaks are visually

identified. Surprisingly, the authors only described 2 of the 4 peaks they find with the model. I recommend they discuss the dynamics of the protein-like component. Given the results presented, it would be informative to see how well this component correlated to other PARAFAC components and in cross section across the lake (ie, as in Figs 2 and 3). Also correlation of this peak with $\delta 13C$ values.

-> We added more details and figures about the fluorescent components identified with the model in the revised version. Since $\delta 13C$ values fall into a narrow range (marine source), no correlation was found for the FDOM components.

10. L140: Range of values does not capture the most negative value reported (–27.8‰).

-> The DOC–$\delta 13C$ values ranged from –22.6‰ to –27.8‰ changed as suggested in the revised version.

11. Discussion L155: How is "significant excess" being defined? It is unclear what the authors mean by this phrase and how they quantified it.

-> specified in the revised version ($\sim$75% higher than the mixing line).

12. L158: What does land-seawater interaction mean? Mixing? Proportional mixing would not add an excess of DOC; an excess implies production in spite of mixing... unless a 3rd source is implied. In this case, binary mixing analysis won't work. Perhaps the authors should suggest here the benthos as a potential source; but that source also should be parameterized (eg, what is its $\delta 13C$-DOC values, FDOMH, FDOMM values, SR etc.).

-> We suggest that "land-seawater interaction" is due to the tidal inundation of seawater on the reclaimed land. This process can cause increases in DOC with depleted DOC–$\delta 13C$ values, high SR values, and non-fluorescent, without salinity decreases. This is more clearly explained in the revised version.

13. L180: The groupings appear arbitrary; what criteria were used to separate them? I

don't understand how the terrestrial source of DOM can be not fluorescent, give that the authors identify humic fluorescence as a specific marker. This section of the discussion is extremely hard to follow.

-> As mentioned above, the DOC in the reclaimed land could be non-fluorescent as it is exposed to sunlight for a long period of time. Otherwise, please suggest alternative explanation for our observed trend. Since this paper is for the observed results, we only can suggest the most plausible mechanism. The groups (1 and 2) were separated based on its DOC concentrations, DOC–$\delta$13C values, and salinity. We added the details in the revised version.

14. L197: No evidence is provided for photochemical or bacterial degradation in this study.

-> Based on DOC–$\delta$13C values, high SR values, low FDOM and NH4+ concentrations, we suggest the most plausible sources.

15. L201: As suggested earlier, the possibility to use EMMA or other multivariate means with these data are encouraging. I recommend the authors try to analyze their results with an aim of using exploratory methods (eg. Ordination such as PCA or non-parametric techniques) and perhaps 2-way analyses wherein the difference or season (or stream flow if available; not presented) is considered. A clearer way of quantifying the Groups (1 and 2) must be presented at the very least, so that readers can follow the study.

-> PCA or other statistical techniques are useful in differentiating various sources. However, in this study, the excess DOC occurred different locations (low salinity water, high salinity water, and near benthic water). So, we simply try to determine the source of the excess in each sample group.

16. L210: No analysis was presented to demonstrate the linkage of $\delta$13C values and NH4+ values.

-> A previous study (Kim and Kim, 2018) suggested the anaerobic benthic production of FDOMH in low salinity water in this region based on NH4+ concentrations. In this study, we support this finding based on our NH4+ relationships (NH4+ versus DOC and FDOMH correlations) and DOC-$\delta$13C values (marine source). We showed that the source is not due to terrestrial inputs! This is clarified in the revised version.

References

Kim, J. and Kim, T.-H.: Distribution of humic fluorescent dissolved organic matter in lake Shihwa: the role of the redox condition, Estuar. Coast., https://doi.org/10.1007/s12237-018-00491-0, 2018.

Lang, S. Q., Lilley, M. D., and Hedge, J. I.: A method to measure the isotopic (13C) composition of dissolved organic carbon using a high temperature combustion instrument, Mar. Chem., 103, 318–326, 2007.

Panetta, R. J., Ibrahim, M., and Gélinas, Y.: Coupling a high-temperature catalytic oxidation total organic carbon analyzer to an isotope ratio mass spectrometer to measure natural-abundance $\delta$13C-dissolved organic carbon in marine and freshwater samples, Anal. Chem., 80, 5232–5239, https://doi.org/10.1021/ac702641z, 2008.

---

## Author Response (AR1)

**Editor's comment**
I went through both your MS and your interactive responses to the review. I encourage you to submit a thoroughly revised MS by carefully considering these reviews. In particular, I urge you to well frame your case studies into a broader scope in the mechanism of DOM sources and their characterization. Also, you need elucidate the methods section and make your statement well grounded. Finally, as the reviewer pointed out, the presentation of your paper has to be largely improved. When you submit your revised MS, you need to provide a point-to-point letter explaining how you address the comments and concerns from the reviewers. Your revised MS will be sent out for further reviews.

➔ Thank you for your review and comments concerning our manuscript. As suggested we frame our case studies into a broader scope in identifying the source of DOM in coastal waters. We have added more detailed explanations on the methods and discussion sections with new figures, and all reviewers' comments are carefully addressed in the revised manuscript.

**Reviewer # 1**

General comments:
This is, nothing really novel, but a simple and clear study. The authors used combined concentration, stable carbon isotope and fluorescence measurements to characterize the sources of dissolved organic carbon (DOC) in a small coastal bay, the Sihwa Lake in South Korea. The manuscript is well written and the results were clearly described and discussed. I support its publication and wish the following suggestions can be considered.

➔ Thank you for your valuable comment. All your comments are carefully taken into account in the revised version.

Specific comments:
1. The studied Sihwa Lake is a very shallow (<8 m) coastal bay and it appears that the freshwater influence from the few streams to the bay is insignificant from the salinity and DOC-$\delta^{13}$C distributions, only in the land-ocean interface. Therefore, the sediment resuspension could be an important factor influence the DOC concentrations in the bay. The high DOC (or excess DOC) observed in the bottom water in the nearshore stations (only 2–3 meters deep) in 2017 could be influenced by the sediment resuspension or disturbance of the surface sediment, resulting in high sediment porewater DOC fluxed into the bottom water. Usually, the concentrations of porewater DOC in the coastal sediments are much higher than that of the water column. I think the authors mentioned this but more discussion will be good.

➔ We agree that the excess DOC could be from the sediment re-suspension or pore-water exchange in the nearshore stations. We add more in-depth discussion on this based on DOC-$\delta^{13}$C values (average: –22.1‰) in these samples and references in the revised version (lines 212–215).

2. Line 61–62: "The total volume of the Sihwa Lake water is ~$3.3\times10^8$ $m^3$ $y^{-1}$ and the discharge rate is approximately $3.4\times10^8$ $m^3$ $y^{-1}$ (Lee et al., 2014)". Please check on the unit of the total volume.

➔ Yes, corrected. Thank you!

3. For Figure 4 and 5, if these lines are linear regression of the date, the regression parameter should be given.

➔ We add the statistical information ($r^2$ and p values) in the revised version (Figure 4).

4. In all figure captions, Lake should be added after Sihwa.

➔Yes, add "lake" as suggested in the revised version.

**Reviewer # 2**

General comments:
Han et al provide a short summary of DOM properties in Sihwa Lake, a constructed coastal lake in a heavily industrialized coastal area, over 2 sampling trips taken in spring 2017 and in lake summer 2018. Same sites were visited in each sampling. Using a combination of nutrients and optical and stable isotope tracers, they aim to distinguish multiple sources of DOM (though the sources are not clearly identified). The brevity of this manuscript makes it very difficult to follow. Many details are lacking and some deeper analysis is required to support the conclusions made in this study. Several conclusive statements are made without a clear logical argument to help the reader reach the same conclusion. These problems occur throughout this version of the manuscript, and, along with some substantial editing for grammar and usage, require more than substantial revision.

➔ Thank you for your review and comments. In this study, we focused on determining the sources of excess DOC occurring in this bay. Although we cannot elucidate the exact sources and processes in this study, it is

clear that our approach (using DOC–$\delta^{13}$C and optical properties) suggests different sources that cannot be identified with the traditional methods. We showed that the excess DOC dependent on salinity is from marine sources (although it is generally regarded as terrestrial sources), and the excess DOC in the high-salinity water is from terrestrial sources (although it is believed to be from marine sources without our approach). The revised version was thoroughly checked for grammar and usage by a native editor.

Specific comments:

1. L55: Finish the set up for this manuscript. What are the sources expected? It is curious why the authors didn't try to use endmember mixing analysis (EMMA) to disentangle the sources. The primary sources appear to be: terrestrial, marine, phytoplankton, and "anaerobic benthic processes" which I shorten to benthic.
➔ The end-member mixing analysis is very useful for tracing different water-mass mixings. However, the excess DOC occurring in this study is either from the sediment or land as the DOC is directly introduced to low-salinity water or seawater. Thus, it is impossible to do EMMA.

2. Methods L68: It appears the sluice gates are mostly closed; what does periodic opening entail? Were the gates opened prior to sampling?
➔ The sluice gates are opened twice a day (every low tide and high tide). We added more details in the revised version (lines 65–67).

3. L70: What vessel was used for sampling? "a ship" is nebulous.
➔ It is a small boat (~1 ton). mentioned in the revised version (lines 72–74).

4. L78: How many mL of 6M HCl were used and what was the final pH?
➔ We added 20 μL of 6M HCl to each sample. The final pH of the sample was ~2. Details are described in the revised version (lines 79–81).

5. L88: Unlikely that the precision of the TOC analyzer for DSR measurement is 2.2 μM, round to 2 μM. How many analyses?
➔ corrected as suggested in the revised version (line 90). We measured DSR three times per each 10-sample batch.

6. L92: To my knowledge no consensus value of DSR is reported, though similar values have been reported as described here. Reword to indicate this (as was done in earlier work referenced here). If a consensus value is now published, please cite the publication. Also appear report number of analyses (N) for these standards.
➔ The $\delta^{13}$C values for the DSR were reported by Panetta et al. (2008) (–21.37±0.33‰) and Lang et al. (2007) (–21.9±1.3‰). We add more references and details in the revised version (lines 92–97).

7. Results: Make the colorbar ranges for Figs 2 and 3 the same for each panel for ease of comparison.
➔ changed as suggested in the revised version (Figure 2 and 3).

8. The PARAFAC results should be tested against the OpenFluor database.
➔ Yes, the PARAFAC results are compared with the OpenFluor database. We added the results in the revised version (lines 107–111).

9. Here, spectral components are compared to Coble 2007 wherein peaks are visually identified. Surprisingly, the authors only described 2 of the 4 peaks they find with the model. I recommend they discuss the dynamics of the protein-like component. Given the results presented, it would be informative to see how well this component correlated to other PARAFAC components and in cross section across the lake (ie, as in Figs 2 and 3). Also correlation of this peak with $\delta^{13}$C values.
➔ We added more details and figures about the fluorescent components identified with the model in the revised version (lines 113–119, 152–157, 163–167; Figure 3). Since $\delta^{13}$C values fall into a narrow range (marine source), no correlation was found for the FDOM components.

10. L140: Range of values does not capture the most negative value reported (–27.8‰).
➔ The DOC–$\delta^{13}$C values ranged from –22.6 to –27.8‰. changed as suggested in the revised version (lines 217–219).

11. Discussion L155: How is "significant excess" being defined? It is unclear what the authors mean by this phrase and how they quantified it.
➔ specified in the revised version (~75% higher than the mixing line) (lines 204–206).

12. L158: What does land-seawater interaction mean? Mixing? Proportional mixing would not add an excess of DOC; an excess implies production in spite of mixing… unless a 3$^{rd}$ source is implied. In this case, binary

mixing analysis won't work. Perhaps the authors should suggest here the benthos as a potential source; but that source also should be parameterized (eg, what is its $\delta^{13}$C-DOC values, FDOM$_H$, FDOM$_M$ values, S$_R$ etc.).

➔ We suggest that "land-seawater interaction" is due to the tidal inundation of seawater on the reclaimed land. This process can cause increases in DOC with depleted DOC–$\delta^{13}$C values, high $S_R$ values, and non-fluorescent, without salinity decreases. This is more clearly explained in the revised version (lines 217–226).

13. L180: The groupings appear arbitrary; what criteria were used to separate them? I don't understand how the terrestrial source of DOM can be not fluorescent, give that the authors identify humic fluorescence as a specific marker. This section of the discussion is extremely hard to follow.

➔ As mentioned above, the DOC in the reclaimed land could be non-fluorescent as it is exposed to sunlight for a long period of time. Otherwise, please suggest alternative explanation for our observed trend. Since this paper is for the observed results, we only can suggest the most plausible mechanism. The groups (1 and 2) were separated based on its DOC concentrations, DOC–$\delta^{13}$C values, and salinity. We added the details in the revised version (lines 200–204).

14. L197: No evidence is provided for photochemical or bacterial degradation in this study.

➔ Based on DOC–$\delta^{13}$C values, high $S_R$ values, low FDOM and NH$_4^+$ concentrations, we suggest the most plausible sources.

15. L201: As suggested earlier, the possibility to use EMMA or other multivariate means with these data are encouraging. I recommend the authors try to analyze their results with an aim of using exploratory methods (eg. Ordination such as PCA or non-parametric techniques) and perhaps 2-way analyses wherein the difference or season (or stream flow if available; not presented) is considered. A clearer way of quantifying the Groups (1 and 2) must be presented at the very least, so that readers can follow the study.

➔ PCA or other statistical techniques are useful in differentiating various sources. We performed PCA for data analyses, but the PCA component result is insignificant to quantify the groups (Fig. right). Thus, we didn't add this result in the revised manuscript. In this study, the excess DOC occurred different locations (low salinity water, high salinity water, and near benthic water). So, we simply try to determine the source of the excess in each sample group separated by DOC concentrations, DOC–$\delta^{13}$C values, and salinities.

[Figure]

16. L210: No analysis was presented to demonstrate the linkage of $\delta^{13}$C values and NH$_4^+$ values.

➔ A previous study (Kim and Kim, 2018) suggested the anaerobic benthic production of FDOM$_H$ in low salinity water in this region based on NH$_4^+$ concentrations. In this study, we support this finding based on our NH$_4^+$ relationships (NH$_4^+$ versus DOC and FDOM$_H$ correlations) and DOC-$\delta^{13}$C values (marine source). We showed that the source is not due to terrestrial inputs! This is clarified in the revised version (lines 191–198).

**References**

[revised manuscript text omitted]

Heejun Han 26/10/20 1:27 PM

Heejun Han 26/10/20 1:28 PM

Heejun Han 26/10/20 1:28 PM

---

## Referee Report (RR1)

This revised manuscript entitled "Characterizing the origin of excess dissolved organic carbon in coastal seawater using stable carbon isotope and light absorption characteristics" by Han, Hwang and Kim has been improved significantly, especially the Discussion. The comments and suggestions made by myself to the authors in the first-round review have been well addressed. I recommend this paper to be published in Biogeochemistry.

Line 338: Please add the publication year for the referenced paper "Stedmon, C.A. and Bro, R.".

---

## Author Response (AR2)

**Reviewer # 1**

This revised manuscript entitled "Characterizing the origin of excess dissolved organic carbon in coastal seawater using stable carbon isotope and light absorption characteristics" by Han, Hwang and Kim has been improved significantly, especially the Discussion. The comments and suggestions made by myself to the authors in the first-round review have been well addressed. I recommend this paper to be published in Biogeosciences.

➔ Thank you for your valuable comments.

Minor comment:
Line 338: Please add the publication year for the referenced paper "Stedmon, C. A. and Bro, R.".

➔ changed as suggested (lines 325–326)

**Reviewer # 2**

The work by Han et al. describes the DOM distributions in Sihwa Lake in spring 2017 and 2018. The multiple properties of DOM have been investigated e.g. the FDOM, the DO13C and so on. I generally agree with the comments by reviewer 1 and 2 in that this work provides a comprehensive view of the characteristics of coastal DOM with the involvement of multiple analysis on different aspects of DOM properties, and the authors have responded the comments in details. However, as pointed out by reviewer 2, the lack of clear logical argument gave me some difficulty to reach the conclusion. This was not quite improved in the revised version submitted. Thus I would suggest the authors to reorganize the discussion section for a more logical argument. The following comments may be of some help for this purpose.

➔ Thank you for your valuable comments. We have thoroughly reorganized the discussion section for better logical flow and clarity.

1. The authors should more clearly summarize the new information this work brings in for our knowledge of DOM cycling in Sihwa Lake. This should be significantly beyond Kim and Kim's work (2016) which studied the optical properties of DOM in Sihwa lake and found the contribution of the porewater to the DOM in water column. What is the major difference between Kim and Kim's finding and the conclusion in this manuscript that sediment OM is the major source of DOM in Sihwa Lake? The introduction section should more focus on the specific processes or new information this work identified. Progresses from other labs should be discussed in details in the introduction. A more logical argument of the conclusion should consider the followings:

➔ In the revised manuscript, all these comments were carefully taken into account (lines 175–181).

2. What is the excess DOM? It is denoted in the manuscript as the part of DOC measured in 2017 above the mixing line of DOC in 2018. If my understanding is correct, this idea tries to evaluate the DOM influenced by conservative mixing of two endmembers (freshwater and seawater DOM) and the part of DOM added / removed during the mixing. However, the excess DOC denoted here is under the assumption that the DOC concentrations of freshwater endmembers remained constant between 2017 and 2018. Please provide enough evidence and argument to prove this assumption is reasonable. Plus, the excess DOM should be denoted at the first place it is mentioned except in the abstract.

➔ We agree with the reviewer that the terrestrial input can be temporally variable. We decided to use the DOC concentrations in incoming open-ocean water as a background and considered any values higher than the background value as an excess (lines 193–195). Therefore, in this case, additional riverine input becomes a part of the excess DOC. The sources of excess DOC were discriminated as follows: (1) marine sediment source in low-salinity waters, (2) terrestrial source in some high-salinity stations, and (3) marine source in some high-salinity stations.

➔ In addition, all other comments were taken into account in the revision.

3. Clarify the sources based on the properties of DOM. The origin of DOM in group 2 in the conclusion (and in the abstract) is more like a description of DOM properties which are characterized as lower DO13C, non-fluorescent, and low MW. The author should clarify what is the specific source and why the DOM from this source have these properties. Clarify the difference of terrestrial OM in group 2 from the source of DOM in the low salinity water.

➔ We have added some more details on this issue in the discussion (lines 211–214).

Minor comment:
Some minor comments as following:
1) whenever use "significant" in the text, please provide statistical analysis;
2) FDOMH mentioned in discussion was not defined in method section;
3) Change "uM" to "umolL-1"

4) Line 31: delete "reduced".

➔ All minor comments were taken into account in the revised version.

**Reviewer # 3**

1. Please provide background information on source and dynamics of $NH_4^+$ in coastal waters in the introduction section before using it as a tracer in the manuscript.

➔ We used $NH_4^+$ to show the general environmental condition of study sites, and thus we refrained from providing detailed description on $NH_4^+$ cycle in the introduction.

2. DO is measured in this study, but not used or discussed. It can be taken out if it does not add any useful information.

➔ We added a brief description on DO in the revised version (lines 177–180). Also, DO is usually a critical parameter in biogeochemistry and hence we wanted to show this parameter although it was not described as a major controlling factor of DOM properties in our study.

3. Please provide water residence time in spring vs. fall if possible.

➔ It is not easy to know the exact residence time of the water. The residence time of the water body in the lake is likely controlled mainly by water exchange through the sluice. We, therefore, provide a rough estimation of the turnover time of the water to be about one year simply based on the water exchange rate and the volume of the lake water.

4. The DOC input source may vary significantly with seasons, so it might not be appropriate to used the mixing line based on data from September of 2018 as the baseline to define the excess DOC for data from March of 2017. A proper assumption has to be made.

➔ Yes, we changed this in the revised version. We decided to use the DOC concentrations in incoming open-ocean water as a background for calculating the excess. Please see our response to Reviewer #2.

5. The authors suggested the marine sediments as the major source of DOC, based on carbon isotopic signature and correlation between DOC and $NH_4^+$, which seems to be rational. But a linear mixing line showing a decreasing trend of DOC with salinity was also observed. This usually occurs when terrestrial input is the major source of DOC. Please explain why marine sediments as the major source could result in such a DOC distribution pattern.

➔ Although the DOC and salinity showed linear correlation indicating possible contributions of terrestrial inputs, our DOC–$\delta^{13}C$ values (–20.0±0.4‰) observed in low-salinity (<28) waters here exclude possible significant contributions of terrestrial sources, indicating the effective degradation of terrestrial sources before they reached the mixing zone. In this kind of tidal flat environments, the large inputs (i.e., seepage) of DOC and $FDOM_H$, together with $NH_4^+$, could happen from marine sediments, as the shore water runs back and forth on a wide sediment-surface area over a tidal cycle (Kim et al., 2012). In this case, depending on salinity ranges of overlying waters, different slopes of DOC and $FDOM_H$ against salinities could be observed for different seasons as shown in this study (lines 175–181).

Minor comment:
Line 290, please clarify how bottom sediment porewater was excluded as the source of DOC.
Line 299, changes in input sources could also result in $S_R$ variations.
Figure 2, please provide information on how grouping was defined in the caption.
Figure 5b, I suggest drawing a single regression line instead of two.

➔ We made all above technical corrections as suggested in the revised version.

**Reference:**

Kim, T.-H., Waska, H., Kwon, E., Suryaputra, G. N., and Kim, G.: Production, degradation, and flux of dissolved organic matter in the subterranean estuary of a large tidal flat, Mar. Chem., 142–144, 1–10, http://dx.doi.org/10.1016/j.marchem.2012.08.002, 2012.

---

## Author Response (AR3)

Dear authors,

I read through your further revised MS and see the MS still needs some extent of revisions:

1) The claimed excess DOC is not supported by your data and I urge you to make proper scientific justifications or largely tone down your statement as per the sources of excess DOC.

2) I strongly suggest you have your further revised MS proof-read by a native English speaker.

Dear Editor,

Thank you for the review and valuable comments on our manuscript. In this study, we define that "the excess DOC represents any DOC concentrations higher than those in the incoming open-ocean seawater" (lines 225–229). If the DOC in fresh water is mainly from streams (terrestrial origins), we could easily define the excess DOC as any DOC concentrations higher than the linear mixing concentration between the fresh water and open ocean water (the common approach). However, we reveal that the DOC in fresh water is mainly from sediments (tidal flat), which should be regarded as excess inputs, in this region. Thus, this definition could be confused, and thus we further clarified this in the revised version. In addition, we carefully edited the manuscript by a native English speaker.

[revised manuscript text omitted]